# Combining Whole Genome Sequencing Data from Human and Non-Human Sources: Tackling *Listeria monocytogenes* Outbreaks

**DOI:** 10.3390/microorganisms11112617

**Published:** 2023-10-24

**Authors:** Ingrid H. M. Friesema, Charlotte C. Verbart, Menno van der Voort, Joost Stassen, Maren I. Lanzl, Coen van der Weijden, Ife A. Slegers-Fitz-James, Eelco Franz

**Affiliations:** 1Centre for Infectious Disease Control, National Institute for Public Health and the Environment (RIVM), 3721 MA Bilthoven, The Netherlands; maren.lanzl@rivm.nl (M.I.L.); eelco.franz@rivm.nl (E.F.); 2Netherlands Food and Consumer Products Safety Authority (NVWA), 3511 GG Utrecht, The Netherlands; c.c.verbart@nvwa.nl (C.C.V.); c.vanderweijden@nvwa.nl (C.v.d.W.); i.a.fitz-james@nvwa.nl (I.A.S.-F.-J.); 3Wageningen Food Safety Research (WFSR), 6708 WB Wageningen, The Netherlands; menno.vandervoort@wur.nl (M.v.d.V.); joost.stassen@wur.nl (J.S.)

**Keywords:** *Listeria monocytogenes*, human, food, source, cluster, persistence, cgMLST, outbreak investigation, one health, surveillance

## Abstract

*Listeria monocytogenes* (*Lm*) is ubiquitous in nature and known for its ability to contaminate foods during production processes. Near real-time monitoring of whole genome sequences from food and human isolates, complemented with epidemiological data, has been used in the Netherlands since 2019 to increase the speed and success rate of source finding in the case of (active) clusters. Nine clusters with 4 to 19 human cases investigated between January 2019 and May 2023 are described. Fish production sites were most often linked to outbreaks of listeriosis (six clusters), though other types of food businesses can face similar *Lm* problems, as the production processes and procedures determine risk. The results showed that low levels of *Lm* in food samples can still be linked to disease. Therefore, the investigation of a cluster of cases and deployment of the precautionary principle helps to focus on safe food and to prevent further cases. Good practice of environmental monitoring within a food business allows early detection of potential issues with food safety and helps food businesses to take appropriate measures such as cleaning to prevent regrowth of *Lm* and thus future outbreaks.

## 1. Introduction

*Listeria monocytogenes* (*Lm*) is a bacterium that is widespread in the environment; it can be found in soil and water as well as in the feces of humans and animals [1,2]. Food production chains and thus food can become contaminated via asymptomatic fecal shedding and environmental contamination. Besides its opportunistic nature, *L. monocytogenes* is able to survive and even multiply in a broad spectrum of conditions, such as low temperatures, a wide pH range, and relatively low water activity, and it can also form biofilms and endure disinfectants [3,4,5]. Human illness caused by *Lm* is almost always foodborne [6]. In most cases, infection with *Lm* will remain subclinical or cause febrile gastroenteritis [7]. In immunocompromised adults or the elderly, infection can lead to invasive diseases such as bacteremia, septicemia, meningitis, encephalitis, and endocarditis [7,8,9]. Infection during pregnancy can result in abortion, stillbirth, and neonatal listeriosis. Invasive listeriosis is relatively rare compared to other foodborne diseases, but it has one of the highest hospitalization and case fatality rates [10].

Due to the characteristics of *Lm*, it can become a persistent resident in food-processing facilities, which may lead to regular contamination of food products [11]. Spots that are difficult to clean are prone to colonization, creating a persistent source for the contamination of food. Consumption of food products contaminated during processing, which are not heated or do not undergo other steps to control the pathogen before consumption, can lead to listeriosis, especially in vulnerable groups.

The discriminatory power, universality, and robustness of whole genome sequencing (WGS) makes it an invaluable tool for surveillance, cluster detection, and outbreak investigation [12,13].

In the Netherlands, WGS data on *Lm* from food monitoring programs governed by the Netherlands Food and Consumer Product Safety Authority (NVWA) and analyzed by Wageningen Food Safety Research (WFSR) and sequences of *Lm* from patients obtained within the context of the listeriosis surveillance coordinated by the Centre for Infectious Disease Control of the National Institute for Public Health and the Environment (RIVM) are mutually shared, with the aim of increasing the speed and success rate of source finding in the case of (active) clusters. WGS analyses showed a notable number of clusters with *Lm* sequences to be (re)occurring over multiple years [14]. Regularly, clusters are identified with both human and non-human isolates, upon which the NVWA acts in its risk-based supervision task to verify business’ compliance with regard to controlling *Listeria* in the food production chain. In this paper, the added value of combining shared WGS results with epidemiological data is described, both in response to clusters of listeriosis as well as retrospectively identifying the common source of clusters with both human and non-human isolates.

## 2. Materials and Methods

### 2.1. Listeriosis Surveillance

Listeriosis became a notifiable disease in the Netherlands in December 2008. Laboratory-confirmed cases are required to be reported to the regional public health services by both the physician and the laboratory. The regional public health service contacts the cases or their relatives to complete a short questionnaire about underlying (chronic) diseases, the clinical course of listeriosis, and exposure to possible risk factors, mainly food, in the 30 days before disease onset. This information, together with patient characteristics, is then reported in a national online infectious disease notification database (called Osiris). A mother and her newborn with listeriosis are notified separately, but with reference to each other, and are considered as one case.

Clinical microbiological laboratories can voluntarily send in *Lm* isolates to the Netherlands Reference Laboratory for Bacterial Meningitis (NRLBM), which forwards the isolates to the National Institute for Public Health and the Environment (RIVM) for further typing for national infectious disease surveillance purposes. WGS became the standard typing method in 2017. Isolates of 2010–2015 were sequenced within the ELiTE study (European *Listeria* Typing Exercise Extension to Whole Genome Sequencing) led by the European Centre for Disease Prevention and Control (ECDC) [15]. To obtain the WGS data of all isolates since 2010, isolates of 2016 were also retrospectively sequenced. Sequencing from 2016 to 2019 was performed by a commercial sequencing company, using Illumina HiSeq (2 × 100 bp) and Illumina NovaSeq (2 × 150 bp), and from 2020 onwards by RIVM, using Illumina NextSeq (2 × 150 bp) (Illumina, San Diego, CA, USA).

### 2.2. Monitoring of Food

The NVWA, as the Competent Authority, enforces food business operators’ compliance with relevant food safety regulations by, among others, conducting food surveillance. The non-human isolates are found through sampling of food products throughout the food chain by food monitoring projects carried out by the NVWA. The NVWA also takes environmental samples in food processing plants during outbreak investigations. Testing of these food products is carried out by WFSR. The analytical results of food samples are checked for compliance with the applicable legislation. The General Food Law (EC no. 178/2002) requires that food cannot be placed on the market if it is unsafe, and a specific regulation (EC no. 2073/2005) on microbiological criteria indicates which specific criteria food must meet as a minimum. In regulation EC no. 2073/2005, a microbiological criterium is set for *Lm*, whereby the concentration of *Lm* may not exceed 100 colony-forming units per gram for the duration of the shelf life of ready-to-eat foods. Food business operators must be able to demonstrate that *Lm* cannot exceed this value for the duration of the shelf life of a product. For ready-to-eat foods able to support the growth of *Lm* and ready-to-eat foods intended for infants or special medical purposes, the microbiological criterium is set to no detection in 25 g of product.

WGS is the standard typing method used at WFSR for the monitoring of *Lm* in food since 2017. Isolates from 2015 and 2016 have also been retrospectively typed using WGS. When *Lm* is detected using ISO 11290–equivalent methods [16,17], single colonies are isolated from the plates used in this method and verified as *Lm* using a MALDI Biotyper. Verified *Lm* isolates are stored as glycerol stocks at −80 °C. Isolates have been routinely added to the stock collection since 2005, creating a collection that can be used for retrospective analyses. DNA extracts of isolates are sent to a commercial sequencing provider in batches. Since 2017, sequencing has been performed on Illumina NextSeq, Illumina MiSeq, and Illumina NovaSeq devices, all at 2 × 150 bp.

### 2.3. Combined Database and Cluster Detection

The sequences (in the form of fastq files) of the isolates are exchanged between WFSR and RIVM, at a monthly to 6-week frequency. At RIVM, the sequences are de novo assembled using an in-house-developed pipeline [18]. Assessment of the core genome multilocus sequence typing (cgMLST; [19]) is done using Ridom SeqSphere+ version 5.0.0 (Ridom GmbH, Münster, Germany). Clusters of sequences were defined based on single-linkage hierarchical clustering, with a threshold of a maximum of seven alleles difference out of the 1701 loci [19]. Since 2019, every newly detected cluster consisting of two or more human isolates with at least one isolate from another source is actively communicated to NVWA by WFSR and RIVM as well as new isolates within an existing joint cluster. Joint clusters with one or two patients or with positive food products from different sectors are in the first instance recorded and investigated when growing. Only cases occurring from 2017 onwards and with less than two years between consecutive cases are counted as cluster-related cases of the described sequence clusters. Sequences from human cases and food up to May 2023 were included in the analyses and description. Data and information gathered from the human cases and production locations were used to examine the likelihood of an epidemiological link between the human cases and food product(s) in the sequence cluster.

## 3. Results

### 3.1. Listeriosis Surveillance and Cluster Detection

A total of 1215 human cases of listeriosis were reported between January 2010 and May 2023, with a range of 72 to 117 cases per year. The majority of these cases (72.5%) occurred in those 65 years of age or older, and 74 cases (6.1%) were pregnancy-related, which resulted in 24 miscarriages/stillbirths/neonatal deaths. Among the adults, 141 deaths (11.6%) were recorded. For 972 cases (80.0%), an isolate for further typing was received. The combined databank further contained 2328 *Lm* isolates from food and environmental samples, collected between 2015 and May 2023.

Since 2019, 79 clusters of sequences with at least one patient who became ill between January 2019 and May 2023 were identified in the combined database: 36 clusters consisted of patients only, and 43 were joint clusters with human and non-human isolates. Most joint clusters included only one patient who became ill from January 2019 onwards (*n* = 23; 53%), and eight included two patients (19%). Clusters that included three or four patients were each observed twice. The remaining eight clusters included 5–19 patients. In nine joint clusters, the NVWA started additional investigations at food businesses involved along the food production chain. Five of these clusters were clusters with human cases dating back to 2017 and/or 2018 as well as new cases seen from 2019 onwards. The maximum difference between the individual sequences within these clusters was three alleles (one cluster), six alleles (four clusters), eight alleles (one cluster), ten alleles (one cluster), or 15 alleles (two clusters). In all nine clusters, at least one sequence from a human case matched with at least one non-human sequence at zero alleles difference. The characteristics of, and investigations at, food businesses of the nine clusters are summarized in Table 1.

### 3.2. Epidemiological Evidence

Following mandatory notification of the case by a laboratory or physician, basic information on the consumption of main food products is gathered by the regional public health service. Unfortunately, this information is not always available, as listeriosis is mainly a disease among the elderly, who can become too ill to provide information, and next of kin do not always know what was eaten in the timeframe of the 30 days prior to disease onset. Nevertheless, the basic information available was enough in three out of the nine investigated clusters to observe a clear tendency between reported consumption of particular products and the identified source products. In three other clusters, the identified products were specifically mentioned. Where possible, cases were approached again with questions about details of specific products identified by the WGS match and subsequent investigation, such as brand and purchase location. In this way, for two additional clusters, the food type could be confirmed. In one case, the brand/type pointed towards two producers, of which one was already in the picture because of previous positive samples with the particular strain. In the other cluster, several, although not all, of the purchase locations mentioned by cases could be linked to a production location. In the case of the cluster related to cold cuts of meat, the epidemiological evidence was weak, as the food type implicated remained very general, e.g., at the level of meat consumption. As a variety of cold cuts sold at several large supermarket chains were contaminated, and people who do eat cold cuts generally eat different types within a 30-day period, it was decided to not pursue the cases to ask for additional information about consumption of cold cuts of meat.

### 3.3. Food Products and Production Locations

The food products testing positive for *Lm*, which were almost always well below the limit of 100 colony-forming units per gram (cfu/g), were salmon, mackerel, herring, eel, and shrimps—often these products were smoked versions—and various cold cuts of meat, liverwurst, and goat cheese. Most production locations implicated within these nine investigations were fish processing plants (*n* = 6); in four out of these six instances, more than one fish species was found to be positive. The other locations were meat processing plants (*n* = 2) and a soft cheese producer.

### 3.4. Starting Point of Investigation

The reasons for starting an investigation can differ. Four different starting points were identified for the nine clusters. Two investigations started based upon detecting a new outbreak with only recent cases. In both instances, six cases were seen at the start of the investigation, of which the three or four most recent cases occurred within one month. Isolates from food monitoring that matched the isolates identified in these clusters came at a later moment in time, which helped in solving these two outbreaks. Three existing clusters with cases and non-human isolates before 2019 showed a clear increase in cases, leading to the start of an investigation. Finding a match between human and non-human isolates was the start of the investigations in three additional clusters. And, finally, one investigation started after repeated increasing of the cluster, with a single case each time.

### 3.5. Actions to Locate a Contaminated Production Site

When positive food isolates point towards a certain food production plant as the potential source of the outbreak, the first step the food safety authority takes is to inform a food business. The latter is asked to investigate whether they have recognized indications of a problem in their food safety management system. In one instance, the enterprise had already noticed an increased prevalence of *Lm* found during routine environmental sampling and had recently taken corrective actions based on this information. In the next step, the food safety authority checks whether effective business activities and procedures are in place based on HACCP principles to ensure and verify food safety such as traceability of raw materials and end products, cleaning procedures, and monitoring of the environment and end product. During an inspection of the production location, samples may be taken of food products and/or the environment. When necessary, targeted inspections of hygiene and/or cleaning are carried out. Often, food businesses that have been inspected in these types of outbreak investigations have not yet recognized an increase in the prevalence of *Lm* in their routine environmental and end-product monitoring. Based on the findings, agreements are made with the food business about the measures to be taken.

Environmental swabs taken by the food safety authority during the outbreak investigation tested positive in all nine clusters, although in one outbreak investigation the strains found at the production location did not match the outbreak strain, but instead matched with two other human cases in the combined database. In total, three production locations investigated showed more than one *Lm* strain within the facility in the recent past or at inspection, explaining a total of 19 additional cases notified since 2017. Environmental contamination was detected on food contact materials such as a slicing machine, on a conveyor belt, and in brining solutions, in transportation materials such as crates or carts, in packaging machines, on cleaning materials, in sinks and in floor drains, on areas with condensation, and on walls and floors. If environmental contamination was detected after sampling during an inspection in an outbreak investigation, the situation was re-evaluated, and where appropriate, extra measures were taken. A recall of products was necessary in three of the nine clusters.

In five clusters, the food isolates initially matching with human cases were not conclusive enough to pinpoint one production location as the suspected source of the outbreak. The strain was found on products from different suppliers or on products sampled at the point of sale. In these situations, a trace back was carried out, and all relevant suppliers and production locations were questioned or inspected until the contaminated production location was identified.

### 3.6. Enforcement at Production Level and Follow-Up of Cases

Measures were taken at all implicated production locations, with follow-up testing to determine whether the *Lm* contamination was under control. One production site was temporarily closed for further measures, including thorough cleaning, to control the *Listeria* problem. Within a couple of months of the measures being implemented, the number of new cases matching the cluster isolates markedly decreased, often remaining at zero for a long period. Additional inspections were necessary at five production locations in the period 2019–2023, as new cases emerged that could be linked to these clusters. New cases within a previously detected outbreak cluster may be the result of persistent contamination within a production environment where *Lm* could not be completely eradicated by the food business operators.

The time between the initial outbreak investigation and any additional investigations carried out due to new human cases within a cluster varied between one and three years. The time between the last investigation and the end of May 2023 varied between a few months and almost a year for six clusters. The strains related to the temporarily closed production site have not re-emerged in the combined database, with the last case identified almost four years ago. Similarly, the strains related to two other clusters have not re-emerged for a period of around three years.

## 4. Discussion

Sharing of WGS data has enabled the identification of links between food and human cases. Since classical epidemiological outbreak investigation alone seldomly leads to finding the source of *Lm* clusters (due to small cluster size, severely ill patients, long incubation time, and occurrence of low levels of *Lm* in general foods), this identification of clusters kick-starts source finding. Identifying joint clusters in the period January 2019 to May 2023 and their subsequent investigation led to finding the source of the contaminated food products in nine clusters. Five clusters also included cases with onset of disease in 2017 and 2018, with a total of 111 cases related to the nine clusters. During the investigations, one or more other strains were found in three production locations, which could be related to another 19 cases spread over five strains in two clusters. Thus a total of 130 cases could be attributed to the nine investigated clusters, which is 21% of all notified cases in the period January 2017–May 2023 (*n* = 631 cases).

Overall, almost all food samples found positive in the food monitoring had (very) low levels of *Listeria*, well within the limit of 100 cfu/g. This was also seen in Finland [20]. This is allowed as long as *Lm* growth is limited during shelf life, with the exception of infant formula and food for medical use. However, this also means that the perspective to act is limited for the Competent Authority. The moment cases can be related to *Lm* strains present in a product or production facility, the NVWA can act, regardless of whether the legal limit is exceeded or not, based on the statement in the General Food Law that food should be safe for consumption. Nevertheless, this raises the question of how people become ill when such low levels of *Lm* are detected at the end of production. First, monitoring is based upon sampling. The contamination of the production location and products is not homogeneous, meaning that not every food product will be contaminated and those that are will very likely differ considerably in contamination levels [21]. Second, the bacteria may be able to grow beyond the legal limit during a product’s shelf life than is described in the shelf-life study, followed by potential growth at the consumer phase due to improper storage and/or consumption past the shelf life. This subsequently could raise the levels of *Lm* well beyond the threshold. Finally, *Lm* could be more virulent in cases characterized by underlying disease or advanced age, with some *Lm* strains being fundamentally more virulent, resulting in a higher probability of disease at doses below 100 cfu/g [6,22,23].

The monitoring of food is carried out from primary production up to point of sale. The food products can be sampled at different stages of the production chain and at different production locations. Sometimes, a location with a positive sample can be ruled out as contamination source. In those instances, the NVWA will map the production chain and try to trace back the product. All relevant locations are then inspected, and samples are taken, investigating leads to the source of contamination (i.e., isolation of a strain matching the patient strains). The initial positive food samples from the monitoring were not enough to pinpoint the exact source of contamination in five of the nine clusters, emphasizing that the point in the production chain at which the food product tests positive does not have to be the origin of the contamination. Nevertheless, this information can still indicate that hygienic measures have to be taken or that shelf-life studies should be reviewed at different levels of the production chain due to previously underestimated levels of cross- or post-contamination [24].

When the site where the contamination occurred is identified, the NVWA will make sure that the food business takes appropriate action, ranging from hygienic measures to public warnings and shutting down the production processes until the source of the outbreak has been appropriately dealt with. In most cases, when identifying the production location as the source of an outbreak, it was not possible to immediately identify the exact location and/or spread of the contamination within that production location, nor the specific batches of products that were contaminated and (may) still be on sale or in consumers’ homes. Appropriate actions then need to be taken based on the precautionary principle, which generally leads to more extensive recalls than only those batches that have been identified as contaminated by sampling and could lead to closing of production facilities until the control of *Listeria* is sufficiently shown. On most occasions, hygienic measures—with effective environmental controls—were the designated action. Direct interventions at the production sites seemed to be effective, as a reduction in cases was seen in all clusters. However, most strains reappeared, causing new cases related to the same production location after around one to three years. This reaffirms the characteristic of *Lm* to persist in production locations, the near impossibility of eliminating the bacteria, and the importance of good monitoring and cleaning, although reintroduction of the particular strain via raw materials cannot be ruled out [11,25].

Although some sequence types of *Lm* seem to be associated with specific food categories, it is ubiquitous in nature, without a clear relation to specific reservoir hosts [14]. In addition, cases could be better attributed to the production level than to specific food products [14]. Outbreaks caused by *Lm* have been described with ready-to-eat products, dairy (including cheese), vegetables, fruit, (smoked) fish, and meat as food vehicles [11,20,26,27,28,29]. In the Netherlands, monitoring is conducted on a broad variety of food items, with *Listeria* being found at different prevalence levels in all types of products. Still, matches in the described joint clusters were mainly found in fish products, and to a lesser extent in meat and cheese, but not, for example, in ready-to-eat vegetables or fruit. Several factors could be involved, such as a higher prevalence of *Lm* in fish, differences in the production process of the products, differences relating to the promotion of biofilm formation (i.e., available nutrients, moisture, acidity, salinity), and differences in monitoring procedures.

The source of contamination in the investigated clusters was defined as the production location at which the food product became contaminated. This could be, for example, a slicer, a food-contact surface, a transport crate, or a conveyor belt within the site. Once *Lm* has entered a production location, it often is difficult to eradicate due to biofilm formation and can become persistent [24,29,30]. However, the original source of the *Lm*, which could be, for example, raw materials or, less likely, infected personnel [1,24], and how it was introduced remain unknown in the current investigations.

Besides microbiological (WGS) evidence linking patient isolates to food isolates, it is useful to have additional epidemiological confirmation that food products identified by the WGS match and subsequent investigation were actually consumed by the patients [25,26]. Therefore, information from the cases about food consumption, brands, and purchase locations can help to find and give conclusive evidence of a match of food and human cases in a cluster [20]. Within the *Listeria* surveillance, some obstacles make it difficult to routinely gather detailed information. *Lm* can be present on a wide variety of food products, and, as it has a long incubation period of up to 30 days, food intake over a long period of time needs to be examined. To keep the burden of inquiry within limits for the cases, only basic questions about food consumption are asked. Nevertheless, the reported food products were similar or even identical to the *Lm*-positive product in six clusters. It remains undebated that additional details, such as brands and purchase locations, are of much more help and can establish a clear epidemiological link. Therefore, one should always assess whether it is feasible to try to contact cases again for more details in the case of a cluster. Successfully identifying sources also depends on, amongst others, mapping of the food production chain, sampling of the possible sites, analyzing and typing of the samples, and gathering and utilizing epidemiological data. Investigation of clusters with a smaller number of cases is harder, as the link is often weaker: one or two sporadic cases who match with food samples testing positive, with months or more between the events, do not provide a solid base on which to start an investigation. Nevertheless, this can be a reason to intensify *Lm* control at the relevant food business.

## 5. Conclusions

Overall, (near real-time) cross-sectoral combining of WGS data of isolates from patients and food has proven successful. The eight largest clusters, with 5 to 19 cases, and one of the two clusters with four cases that have been identified since 2019 were successfully traced back to a source. This study shows the potential of contaminated production locations to cause clusters of listeriosis, as these nine clusters accounted for 21% of notified listeriosis cases in the Netherlands. Furthermore, actions taken during an investigation always led to a reduction in cases with that particular strain for a period of at least a few months. Good practice in environmental monitoring and cleaning within a food business can prevent regrowth of *Lm* in relevant places, allowing early detection of potential issues with food safety and the opportunity for food businesses to take appropriate measures to prevent future outbreaks. Fish production sites were most often linked to clusters of *Lm* causing outbreaks of listeriosis. Nevertheless, as the production process and procedures are the main factor, other types of food businesses can face similar *Lm* problems. This study showed that low levels of *Lm* in food samples can result in disease. Therefore, investigation of a cluster of cases and deployment of the precautionary principle helps to focus on safe food and to prevent further cases.

## Figures and Tables

**Table 1 microorganisms-11-02617-t001:** Overview of the joint clusters investigated.

Food Products with *Lm* Isolate	Type Location	Sequence Type (ST)	Serotype	Number of Human Cases (Deceased)	First Case	Last Case	Starting Point of Investigation
salmon	fish processing plant	7	1/2a; 3a	6 (1)	2018	2020	combined database
smoked trout, salmon, mackerel, herring	fish processing plant	1	4b	15 (6)	2020	2023	combined database
smoked salmon, mackerel, eel	fish processing plant	1	4b	12 (4 *)	2019	2022	combined database
smoked salmon, mackerel, herring	fish processing plant	173	1/2a	15 (2)	2017	2023	5 cases in 1 month
smoked eel	fish processing plant	2	4b	19 (1)	2019	2023	outbreak, last 3 cases within 1 month
salmon, shrimps, herring	fish processing plant	8	1/2a	5 (1)	2020	2022	repeated growth cases
cold cuts of meat	meat processing plant	6	4b	19 (3 **)	2017	2019	10 cases in several months
liverwurst	meat processing plant	37	1/2a	14 (2)	2018	2023	3 cases in 2 months
goat cheese	soft cheese producer	1	4b	6 (0)	2019	2020	outbreak, last 4 cases within 1 month

* of which, one neonatal death; ** and also one miscarriage.

## Data Availability

All data relevant to the study are included in the article. Additional supporting data may be available from the corresponding author on request. These data are not publicly available as they refer to production locations and commercial food businesses.

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
