# Peer review of "Combining Whole Genome Sequencing Data from Human and Non-Human Sources: Tackling Listeria monocytogenes Outbreaks"

_microorganisms, 2023, doi:10.3390/microorganisms11112617_

Round 1
Reviewer 1 Report
In this work, combining whole genome sequencing data from human and non-human sources were investigated for tackling Listeria monocytogenes outbreaks. As the authors mentioned, the combining shared WGS results with epidemiological data is described, both in response to clusters of listeriosis as well as retrospectively identifying the common source of clusters with both human and non-human isolates. Following are some detail comments.
1. Clusters were mentioned throughout the manuscript. What does the work clusters mean in the work? Gene clusters, a group patience infected by LM with same symptom, or outbreak in same location? Please specify in the manuscript. It is not clear.
2. Information in Table 1 is not clear. Are these number of cases (deceased) from human? From the type location, it seems these LM are from food processing plant. How did the authors get the relationship between number of cases and type location. What does the abbreviation ST mean here?
3. Information about the period and location about surveillance of LM in patience and food are missing. Does the survey mainly focus on recent or several years ago? If the survey is about some years ago, how about the statues about LM pollution now?
4. There are very limited information in materials and methods section. How does the LM contaminated samples (human and non-human sources) collected and treated? Does Listeria monocytogenes outbreak still a problem in the Netherlands?
5. There are a mass of cases analyzed in the work. I would suggest the authors should discussed these cases in tables and figures
6. From the results, 1215 human cases were collected in this work. How many non-human sources cases were collected?
Author Response
Thank you for the opportunity to improve our manuscript. Please find below our response per comment.
In this work, combining whole genome sequencing data from human and non-human sources were investigated for tackling Listeria monocytogenes outbreaks. As the authors mentioned, the combining shared WGS results with epidemiological data is described, both in response to clusters of listeriosis as well as retrospectively identifying the common source of clusters with both human and non-human isolates. Following are some detail comments.
- Clusters were mentioned throughout the manuscript. What does the work clusters mean in the work? Gene clusters, a group patience infected by LM with same symptom, or outbreak in same location? Please specify in the manuscript. It is not clear.
Clusters are based upon sequencing only, defined based on single-linkage hierarchical clustering, with a threshold of a maximum of seven alleles difference out of the 1701 loci. This is described in 2.3. Combined database and cluster detection, lines 119-121. Available meta data and epidemiological information were used for describing and analysing the clusters of sequences. We have added to following sentence: “Data and information gathered from the human cases and production locations was used to examine the likelihood of an epidemiological link between the human cases and food product(s) in the sequence cluster.” (lines 129-131)
- Information in Table 1 is not clear. Are these number of cases (deceased) from human? From the type location, it seems these LM are from food processing plant. How did the authors get the relationship between number of cases and type location. What does the abbreviation ST mean here?
The type of location was established based upon tracing the food product that was positive for LM. To make this more clear, we have now placed the food products in the first column. The number of cases are indeed human cases – which we have added to the table – and are the number of cases with a Listeria strain matching the isolate(s) from the food product(s), as the clusters are based upon sequence data. ST is the abbreviation of sequence type, which is now added.
- Information about the period and location about surveillance of LM in patience and food are missing. Does the survey mainly focus on recent or several years ago? If the survey is about some years ago, how about the statues about LM pollution now?
Listeriosis became notifiable in the Netherlands in December 2008, and WGS has been done on all available human isolates from 2010 onwards (see 2.1. Listeriosis surveillance). WGS on food is available for isolates since 2015 (see 2.2. Monitoring of food). Data from human cases and food up to May 2023 were included in the analyses and description. This has been added to 2.3. (lines 128-129). We focussed in the manuscript on recent clusters, with at least one human case between January 2019 and May 2023 (lines 144-158).
- There are very limited information in materials and methods section. How does the LM contaminated samples (human and non-human sources) collected and treated? Does Listeria monocytogenes outbreak still a problem in the Netherlands?
We get the isolates from humans via the Netherlands Reference Laboratory for Bacterial Meningitis (NRLBM), which get the isolates from the local clinical microbiological laboratories. These latter laboratories are the ones that do the testing of the human samples (see 2.1. Listeriosis surveillance). The non-human isolates are found through sampling of food products throughout the food chain by food monitoring projects carried out by the Netherlands Food and Consumer Product Safety Authority (NVWA). The NVWA also takes environmental samples in food processing plants during outbreak investigations. These non-human samples are analyzed by Wageningen Food Safety Research (WFSR). This information is added to 2.2. Monitoring of food (lines 90-93).
- There are a mass of cases analyzed in the work. I would suggest the authors should discussed these cases in tables and figures
The focus of the manuscript is on the nine clusters which were analyzed and handled in a cross-sectoral manner. These clusters are summarized in Table 1. We think that additional tables and figures will not add information, as the paper serves to give a general idea of what is practiced in the Netherlands.
- From the results, 1215 human cases were collected in this work. How many non-human sources cases were collected?
We have added the requested numbers to 3.1. Listeriosis surveillance and cluster detection (lines 141-143): The combined databank further contained 2328 LM isolates from food and environmental samples, collected between 2015 and May 2023.
Reviewer 2 Report
In the manuscript “Combining whole genome sequencing data from human and non-human sources: tackling Listeria monocytogenes outbreaks” the authors describe the benefits of combining shared WGS results with epidemiological data both in response to clusters of listeriosis as well as retrospectively identifying the common source of clusters with both human and non-human isolates.
Please consider these suggestions for improving this manuscript:
1. I suggest authors should describe in more details how WGS benefits the epidemiological analysis of listeriosis and the connection between different human and non-human Lm isolates. More specifically, they should provide examples on some cases of listeriosis and the correlations between the implicated isolated strains.
2. In the Table 1, the column “Food products positive for Lm” describes food products which are characterised by the presence of Lm or by Lm at numbers >100cfu/gr? I believe that the authors should be more accurate on that.
Author Response
Thank you for the opportunity to improve our manuscript. Please find below our response per comment.
In the manuscript “Combining whole genome sequencing data from human and non-human sources: tackling Listeria monocytogenes outbreaks” the authors describe the benefits of combining shared WGS results with epidemiological data both in response to clusters of listeriosis as well as retrospectively identifying the common source of clusters with both human and non-human isolates.
Please consider these suggestions for improving this manuscript:
- I suggest authors should describe in more details how WGS benefits the epidemiological analysis of listeriosis and the connection between different human and non-human Lm isolates. More specifically, they should provide examples on some cases of listeriosis and the correlations between the implicated isolated strains.
In all nine clusters at least one sequence from a human case matched with at least one non-human sequence at zero alleles difference. The maximum difference between the individual sequences within these clusters was three alleles (one cluster), six alleles (four clusters), eight alleles (one cluster), ten alleles (one cluster) or 15 alleles (two clusters). This has been added to 3.1. Listeriosis surveillance and cluster detection (lines 153-157).
- In the Table 1, the column “Food products positive for Lm” describes food products which are characterised by the presence of Lm or by Lm at numbers >100cfu/gr? I believe that the authors should be more accurate on that.
We have changed ‘food product positive for Lm’ into ‘food product with Lm isolate’ in Table 1 to be more clear that it is presence irrespective of the numbers.